# Associations of Sex Steroids and Sex Hormone-Binding Globulin with Non-Alcoholic Fatty Liver Disease: A Population-Based Study and Meta-Analysis

**DOI:** 10.3390/genes13060966

**Published:** 2022-05-27

**Authors:** Xiaofang Zhang, Yuchan Mou, Elif Aribas, Masoud Amiri, Jana Nano, Wichor M. Bramer, Maryam Kavousi, Robert J. de Knegt, Eralda Asllanaj, Mohsen Ghanbari

**Affiliations:** 1Department of Epidemiology, Erasmus MC University Medical Center Rotterdam, 3000 CA Rotterdam, The Netherlands; x.zhang.1@erasmusmc.nl (X.Z.); y.mou@erasmusmc.nl (Y.M.); e.aribas@erasmusmc.nl (E.A.); amiri.masoud@gmail.com (M.A.); m.kavousi@erasmusmc.nl (M.K.); eralda.asllani@gmail.com (E.A.); 2Institute of Epidemiology, Helmholtz Zentrum München, German Research Center for Environmental Health, 85764 Neuherberg, Germany; jana.nano@helmholtz-muenchen.de; 3German Diabetes Center, 85764 München-Neuherberg, Germany; 4Medical Library, Erasmus MC University Medical Center Rotterdam, 3000 CA Rotterdam, The Netherlands; w.bramer@erasmusmc.nl; 5Department of Gastroenterology and Hepatology, Erasmus MC University Medical Center Rotterdam, 3000 CA Rotterdam, The Netherlands; r.deknegt@erasmusmc.nl; 6Department for Health Evidence, Radboud Institute for Health Sciences, Radboud University Medical Center, 6525 EZ Nijmegen, The Netherlands

**Keywords:** gonadal steroid hormones, non-alcoholic fatty liver disease, NAFLD, sex characteristics, meta-analysis

## Abstract

Background: Prior studies have reported inconsistent results or less well-explored associations between sex hormones and non-alcoholic fatty liver disease (NAFLD). Here, we aimed to investigate the associations of NAFLD with sex steroids and sex hormone-binding globulin (SHBG) in the population-based study and conduct a comprehensive systematic review and meta-analysis of all published observational studies. Methods: Analyses included 755 men and 1109 women with available data on sex steroids, SHBG, and ultrasound-based NAFLD from the Rotterdam Study. Multivariable regression models were used to examine the associations. Additionally, we searched five databases from inception to 1 April 2022 and performed a systematic review and meta-analysis. Random-effects (DerSimonian-Laird) method was used for meta-analysis, odds ratios (ORs) were calculated for the effect estimate, subgroup and leave-one-out sensitivity analyses were conducted, and meta-regression was performed to explore the pooled statistics with high heterogeneity. Results: In the Rotterdam Study, lower levels of SHBG were associated with NAFLD in both sexes, while lower testosterone was associated with NAFLD only among women. Similarly, the meta-analysis of 16 studies indicated no sex-specific association between SHBG and NAFLD (men: OR = 0.37, 95%CI 0.21–0.53; women: OR = 0.40, 95%CI 0.21–0.60), yet there was a sex-specific association between testosterone and NAFLD (men: OR = 0.59, 95%CI 0.42–0.76; women: OR = 1.06, 95%CI 0.68–1.44). Moreover, men with NAFLD had lower estradiol levels than those without NAFLD. Conclusions: Lower SHBG levels were associated with NAFLD in both sexes, but testosterone levels were associated in a sex-specific manner. In addition, our results showed estradiol with the potential as a protective factor for NAFLD in healthy men.

## 1. Introduction

Non-alcoholic fatty liver disease (NAFLD) is the most common cause of chronic liver disease worldwide with a substantial health care burden [1]. NAFLD is characterized by the presence of hepatic steatosis, defined by imaging or histology, in individuals who consume little or no alcohol and do not have secondary causes of hepatic fat accumulation [2]. NAFLD has a potentially progressive course leading to non-alcoholic steatohepatitis (NASH), and patients with NASH are at increased risk of liver fibrosis, cirrhosis, and even liver cancer [3,4]. Moreover, emerging evidence has suggested that NAFLD is more common in men than in women [5,6,7,8], and sex differences in the prevalence of liver disease may result from the levels of sex hormones and hepatic expression of sex hormone receptors [9,10,11] Previous studies [12,13,14], however, have reported inconsistent results for the association between sex hormones and NAFLD. The mechanisms underlying these controversial results are not entirely understood, which has led to an interest in exploring the association between sex steroids and NAFLD.

Prior studies indicated that higher levels of testosterone and sex hormone-binding globulin (SHBG) in middle-aged males are associated with a reduced risk of metabolic syndrome [15] and women with type 2 diabetes (T2D) had lower SHBG levels [16]. In addition, estradiol, which is an estrogen steroid hormone, is thought to have complex effects on the liver but is mainly investigated in animal models [17,18]. Little studies have investigated the link between dehydroepiandrosterone (DHEA) or dehydroepiandrosterone sulfate (DHEAS) and metabolic syndrome [15]. A previous study reported that deficiency of DHEAS may play a role in the histologic evolution of the full spectrum of NAFLD [19]. A meta-analysis of 16 studies published until April 2016 (date last searched) has suggested that lower SHBG is associated with NAFLD in both sexes, whereas the association with testosterone was sex-specific [20]. However, this meta-analysis was limited only to testosterone and SHBG in association with NAFLD, included studies with adolescent participants [21], and pooled results for the association of testosterone and SHBG with NAFLD that were reported by each study in an inconsistent way. More importantly, the previous meta-analysis [20] pooled results of studies on fatty liver disease [22] and liver fat [13] with NAFLD. 

The aim of this study was to explore the association of NAFLD with circulating sex steroids (estradiol, testosterone, DHEA, and DHEAS) and SHBG in the prospective population-based Rotterdam Study. Additionally, we performed a comprehensive systematic review and meta-analyzed all published observational studies on sex hormones and NAFLD among adults. 

## 2. Materials and Methods

### 2.1. Analysis of the Rotterdam Study

#### 2.1.1. Study Population

The current study was embedded within the framework of the Rotterdam Study (RS), a prospective cohort study of individuals aged ≥40 years living in the Ommoord district of Rotterdam, the Netherlands. The objectives, design, and methods of the Rotterdam Study have been described in detail elsewhere [23]. For this study, we included participants from the third cohort of the Rotterdam Study (RS-III) (*N* = 3932). We excluded 626 participants from the first visit of the third cohort (RS-III-1) because of missing data for sex steroids and SHBG, leaving 3306 eligible participants. After a median follow-up period of 5.6 years, 816 participants who did not undergo abdominal ultrasound due to diet or moving to other cities, 200 participants who did not provide information about alcohol consumption, and 426 individuals who had excessive alcohol consumption, were stetaogenic drug users or had viral hepatitis were excluded from the study in the second visit of the third cohort (RS-III-2). Finally, 1864 participants (755 men and 1109 women) were included in the analysis (Figure 1). Follow-up examinations were scheduled periodically, approximately every 3–6 years. All participants in the study provided written informed consent to participate and to obtain information from their treating physicians.

#### 2.1.2. Assessment of Non-Alcoholic Fatty Liver Disease

Abdominal ultrasound was performed by a certified and skilled technician using Hitachi HI VISION 900. Images were stored digitally and re-assessed by a single hepatologist with more than 10 years of experience in ultrasonography. Diagnosis of hepatic steatosis was determined dichotomously as the presence of hyperechogenic liver parenchyma according to the protocol by Hamaguchi et al. [24]. NAFLD was defined as without secondary cause for hepatic steatosis such as chronic viral hepatitis and/or excessive alcohol intake [25]. The criteria of metabolic dysfunction-associated fatty liver disease (MAFLD) were based on evidence of hepatic steatosis, along with one of the following three criteria, namely overweight or obesity, presence of T2D, or evidence of metabolic dysregulation [26].

#### 2.1.3. Assessments of Sex Steroids and SHBG

All blood samples were drawn in the morning (≤11 a.m.) after fasting. Total estradiol levels were measured with a COBAS 8000 Modular Analyzer (Roche Diagnostics GmbH). The minimum detection limit for total estradiol was 18.35 pmol/L. Undetectable estradiol was scored as 18.35 pmol/L. Serum SHBG with the Immulite platform (Diagnostics Products Corporation Breda, the Netherlands) and a lower limit detection of the assays was 0.02 nmol/L. Serum levels of total testosterone were measured with liquid chromatography-tandem mass spectrometry (LC-MS/MS) with a corresponding inter-assay of less than 5% and a lower limit of qualification of 0.07 nmol/L. The corresponding inter-assay coefficients of variations (CVs) for total estradiol, SHBG, and total testosterone were <7%, <5%, and <5%, respectively. Serum DHEA and DHEAS were measured on a Waters XEVO-TQ-S system (Waters, Milford, MA, USA) using CHS^TM^ MSMS Steroids Kit (Perkin Elmer, Turku, Finland). The inter-assay CVs of DHEA and DHEAS were <6.5%.

Assessment of covariates and statistical analysis in the Rotterdam Study were described in detail in Appendix A.

### 2.2. Systematic Review and Meta-Analysis

#### 2.2.1. Data Sources and Search Strategy

The study was conducted using a predesigned protocol (which was not registered on online platforms) and following the published guidelines on how to perform a systematic review and meta-analysis [27], and was reported in accordance with PRISMA guidelines (Appendix A). The studies published until 1 April 2022 (date last searched) were searched in five bibliographic databases from inception: Embase.com accessed on 17 April 2022, Medline ALL (Ovid), Web of Science Core Collection, Cochrane Central Register of Trials, and Google scholar accessed on 17 April 2022. The search was performed by an experienced medical information specialist (WMB). In Embase and Medline databases, articles were searched by thesaurus terms, title, and/or abstract; in other databases, only by title and/or abstract. Conference abstracts were removed from Embase, and articles were limited to English-only language and human studies. No limits for the publication date were used. The search combined terms related to the exposure (e.g., sex hormones, estradiol, testosterone, SHBG, DHEA, and DHEAS) and outcome (NAFLD). The full search strategy is provided in Appendix A. The search results were imported in EndNote and duplicated with the method published by Bramer et al. [28].

#### 2.2.2. Eligibility Criteria and Study Selection

Studies were eligible for inclusion if they (i) were cross-sectional, case-control, or cohort studies; (ii) assessed sex hormones (estradiol, testosterone, SHBG, DHEA, and DHEAS); (iii) were conducted with participants aged ≥18 years and without comorbidity of NAFLD; (iv) collected data on NAFLD, and; (v) reported the association of any of the above-mentioned sex hormones with NAFLD. We screened the retrieved titles and/or abstracts and selected eligible studies according to the predefined selection criteria (Appendix A). The full texts of the selected records that met the selection criteria were obtained and examined further by two independent researchers (XZ and MA). In case of disagreement, the decision was made through consensus or consultation with a third independent researcher (EA). Full texts were retrieved for studies that met all the selection criteria. 

#### 2.2.3. Data Extraction and Quality Assessment

Data extraction and quality assessment were independently conducted by two researchers (XZ and EA) using a predesigned data extraction form. Potential bias within each study was evaluated by using the nine-star Newcastle-Ottawa Scale (NOS) [29], a semi-quantitative scale designed to evaluate the quality of case-control or cohort studies. We evaluated cross-sectional studies using an adapted version of the scales downloaded from JBI’s critical appraisal tools assist (https://jbi.global/critical-appraisal-tools (accessed on 1 July 2021)). Study quality was assessed based on these items: the selection criteria of participants, comparability of cases and controls, and exposure and outcome assessments. The NOS assigns a maximum of 4 points for selection, 2 points for comparability, and 3 points for exposure or outcome. Nine points on the NOS reveal the highest study quality [29].

#### 2.2.4. Data Synthesis and Analysis

For the meta-analysis, we used the ORs of the most adjusted models reported by each study. To enable a consistent approach for meta-analysis and interpretation of findings, OR estimates for the associations of sex hormones and NAFLD that were differently reported by each study (such as per-unit or comparing quintiles, quartiles) were transformed, using methods previously described [30]. These transformed estimates consistently corresponded to the comparison of the top versus the bottom of the third of sex hormones distribution in each study. In brief, assuming a normally distributed exposure (e.g., log SHBG) with a log-linear association with NAFLD, conversion factors to convert log ORs from reported scale comparison to the top versus the bottom of the third comparison were derived based on the ratios of expected differences in mean levels of the standardized exposure, for the target comparison versus the reported comparison. Hence, log OR estimates were transformed assuming a normal distribution, with the comparison between the top and bottom thirds being equal to 2.18 times the log OR for per-unit increase (or 2.54 times the log OR for a comparison of extreme quarters, or 2.80 times the log OR for a comparison of extreme quintiles). The method has been used previously in numerous published meta-analyses [31,32]. We retrieved the effect sizes and standard errors for each study and pooled the data. For the main meta-analysis, we used R Package “meta” version 4.18-2 [33] and random-effect models to synthesize effect sizes. Heterogeneity across the included studies was assessed using the Cochran Q statistics and I^2^ statistics, with I^2^ statistics of 25–50%, 50–75%, and >75% considered as mild, moderate, and severe heterogeneity, respectively. For the subgroup meta-analysis, we used the R package “dmetar” version 3.0-2 [34] to evaluate the potential confounding effect of heterogeneity. *p*-value was used to compare the difference between the groups and the value < 0.05 was considered a significant difference. For the leave-one-out sensitivity analysis, we used the R package “metafor”, “dmetar”, and “InfluenceAnalysis” functions to recalculate the results of the meta-analysis N times (N = the number of the included studies for meta-analysis), each time leaving out one study. For the meta-regression analysis, we used R package “metafor” and “multimodel.inference” functions to investigate patterns of heterogeneity in the dataset. Publication bias was evaluated through a funnel plot and Egger’s test. 

## 3. Results

### 3.1. Results of the Rotterdam Study

#### 3.1.1. Characteristics of the Study Population

The demographics, lifestyle factors, medication use, cardiometabolic risks, and sex hormones profiles of men and women in the sample by NAFLD status are shown in Table 1. Among the study participants, 36.2% of men and 32.1% of women had NAFLD. Both men and women with NAFLD had poorer cardiometabolic profiles in terms of BMI, waist circumference, hypertension, T2D, HDL, and triglycerides. Moreover, both men and women with NAFLD had lower total testosterone, SHBG, and DHEA levels than those without NAFLD. Furthermore, 52.2% of women with NAFLD were postmenopausal. The mean age of women with and without NAFLD at menopause was 58.1 ± 4.5 and 57.8 ± 5.1 years, respectively. 

#### 3.1.2. Association of Sex Steroids, SHBG, and NAFLD

Among men (Table 2) in the basic model, lower testosterone levels [highest versus lowest tertile (T3 versus T1): OR = 0.57 (0.37–0.86), *p* trend = 0.036], lower SHBG levels [T3 versus T1: OR = 0.37 (0.24–0.56), *p* trend = 1.56 × 10^−6^], and lower DHEA levels [T3 versus T1: OR = 0.65 (0.42–1.00), *p* trend= 0.043] were associated with NAFLD. Following further adjustment for hypertension, T2D, HDL, and triglycerides in model 2, only lower SHBG levels [T3 versus T1: OR= 0.46 (0.30–0.71), *p* trend = 0.0012] were significantly associated with NAFLD. Furthermore, when we adjusted for cardiometabolic risk factors, the association of total testosterone and DHEA with NAFLD was no longer statistically significant. No associations were found between total estradiol, DHEAS, and NAFLD in any of the models. All associations mentioned above remained significant after applying a strict Bonferroni correction for 5 tests (*p*-value < 0.01), except the association between DHEA and NAFLD among men in the basic model (Table 2).

Among women (Table 3) in the basic model, lower total testosterone levels [T3 versus T1: OR= 0.68 (0.48–0.97), *p* trend = 0.001], and lower SHBG levels [T3 versus T1: OR = 0.27 (0.18–0.38), *p* trend = 2.23 × 10^−16^] were associated with NAFLD. In model 2, lower SHBG levels [T3 versus T1: OR = 0.34 (0.23–0.52), *p* trend = 1.20 × 10^−9^] were associated with NAFLD. There was no association between total estradiol, DHEA, DHEAS, and NAFLD in any of the models. The associations between SHBG levels and NAFLD remained significant after applying a strict Bonferroni correction for 5 tests (*p*-value < 0.01).

In the sensitivity analyses, after adjustment for waist circumference, use of exogenous hormones, or lipid-lowering medication, we did not observe any significant change in the identified associations of sex steroids and SHBG with NAFLD in men (Appendix A) and women (Appendix A). However, the association was changed slightly with additional adjustment for sex hormones simultaneously, BMI, or age categories in both sexes, which might be due to a lower statistical power. Furthermore, the different times since menopause slightly changed the associations of sex hormones and NAFLD in postmenopausal women (Appendix A). Furthermore, we found that 601 out of the 1864 participants had MAFLD, and the association of sex hormones with MAFLD was consistent with NAFLD (Appendix A).

### 3.2. Systematic Review and Meta-Analysis

#### 3.2.1. Literature Search, Characteristics, and Quality of Eligible Studies

As shown in Figure 2, a total of 3658 potentially relevant records were identified from five databases. After deduplication, 2459 references were screened on title and abstract. After screening and detailed full-text assessment, 16 studies [12,14,35,36,37,38,39,40,41,42,43,44,45,46,47] were eligible for the systematic review. The detailed characteristics of these studies (e.g., lead author, publication date, location, study design, adjustment for covariates) are outlined in Table 4.

#### 3.2.2. Pooled Analysis

The meta-analysis for testosterone is based on 10 studies [12,14,35,38,39,40,45,46,47] among men and 7 studies [14,37,40,42,44,46] among women, the findings for SHBG and NAFLD derive from seven studies among men [14,35,41,43,45] and 7 studies among women [14,35,37,41,42,43], the results for estradiol and NAFLD include three studies [36,45] among men. The pooled ORs for NAFLD comparing T3 versus T1 of testosterone were 0.59 (95%CI 0.42–0.76, I^2^ = 93%, *p*-value < 0.01) among men and 1.06 (95%CI 0.68–1.44, I^2^ = 44%, *p*-value = 0.09) among women (Figure 3A). The pooled ORs for NAFLD comparing T3 versus T1 of SHBG were 0.37 (95%CI 0.21–0.53, I^2^ = 49%, *p*-value = 0.07) among men and 0.40 (95%CI 0.21–0.60, I^2^ = 91%, *p*-value < 0.01) among women (Figure 4). The pooled ORs for NAFLD comparing T3 versus T1 of estradiol were 0.90 (95%CI 0.86–0.95, I^2^ = 0%, *p*-value = 0.43) among men (Appendix A).

#### 3.2.3. Subgroup, Leave-One-Out, and Meta-Regression Analyses in Meta-Analysis

We performed a subgroup analysis of included studies for testosterone with NAFLD among men by continents, diagnostic methods, study size, and study design (Table 4). In addition, we performed the leave-one-out sensitivity analysis of included studies of testosterone and NAFLD among men. The biggest changes in the recalculated OR and I^2^ were when leaving out the N. Wang et al. study (OR: 0.55, 95%CI 0.39–0.70, I^2^ = 80%), the details are shown in Appendix A. Moreover, by performing multivariable meta-regression analysis, we found that study size had the highest predictor importance of 12.1% (Figure 3B). The number of included studies on estradiol, SHBG with NAFLD, and testosterone with NAFLD among women was ≤10, and as a general rule of thumb, the subgroup or meta-regression analyses only make sense when the meta-analysis contains at least ten studies (Cochrane Handbook for Systematic Reviews of Interventions Version 6.2, Available online: training.cochrane.org/handbook (accessed on 8 July 2021).

#### 3.2.4. Publication Bias

The appearance of funnel plots was asymmetrical for the analysis of total testosterone and NAFLD in men, and Egger’s test results were significant (*p*-value = 0.011) (Appendix A). This suggests that publication bias may be present. After exclusion of one study [14] that investigated the combined association of other factors and SHBG with NAFLD, findings were not statistically significant (Egger test, *p*-value = 0.51, Appendix A). Since the number of included studies on testosterone and SHBG and estradiol among women was less than ten, which indicates a lack of statistical power, we were not able to perform Egger’s test to check the publication bias [48].

* Quality assessment based on the Newcastle-Ottawa Scale. Range 0 to 9, a higher score is higher quality. Abbreviations: FLD, fatty liver disease; ALD, alcoholic liver disease; NA, not available; NAFLD, non-alcoholic fatty liver disease; NASH, Non-alcoholic steatohepatitis; MAFLD, metabolic dysfunction-associated fatty liver disease; BMI, body mass index; LDL, low-density lipoprotein; TG, triglycerides; HOMA-IR, homeostatic model assessment for insulin resistance; WHR, waist-to-hip ratio; ALP, alkaline phosphatase; DHEAS, testosterone, and dehydroepiandrosterone sulfate levels; HDL, high-density lipoprotein; UA, uric acid; ALT, alanine aminotransferase; MAP, mean arterial pressure; FPG, fasting plasma glucose; γGT, γ-glutamyl transpeptidase; CRP, C-reactive protein; hs-CRP, high-sensitivity C-reactive protein; VAT, variance inflation factors; ESR, erythrocyte sedimentation rate. FBG, fasting blood glucose; HBeAg, hepatitis B e antigen.

## 4. Discussion

In this study, our results showed that people with NAFLD had lower levels of SHBG than people without NAFLD in the Rotterdam Study. The pooled results from the meta-analysis of 16 studies reinforced our finding, suggesting that lower SHBG levels could be associated with NAFLD, however, not in a sex-specific manner. Moreover, men with NAFLD had lower testosterone levels than those without NAFLD, while lower testosterone levels were inversely associated with NAFLD among women. In addition, the pooled results also showed that men with NAFLD had lower estradiol levels than controls. 

### 4.1. Estradiol

Findings from previous studies reported that the incidence and severity of NAFLD increased among older women, particularly postmenopausal women [49,50]. Estradiol has the potential for hepatoprotection by reducing hepatic fat accumulation along with suppressing liver inflammation and fibrosis [51]. Experimental animal studies have reported that supplementation with estradiol protects the liver from various hepatic injuries in male mouse models [52,53]. However, long-term estrogen therapy produces side effects, like headaches, and a small risk of breast cancer in postmenopausal women, and these side effects are rarely investigated in men. In addition, a study that involved 1882 Chinese men showed that estradiol is a protective factor for NAFLD in healthy men [36]. This was consistent with our results from the Rotterdam Study and current meta-analysis. However, our analysis only included three studies on estradiol and NAFLD among men and participants from different ancestors. Therefore, more studies are needed to investigate the association of estradiol with NAFLD, inhibit the side effects, and increase the beneficial effect of estrogen in NAFLD.

### 4.2. Testosterone

To date, it remains unclear which exact molecular mechanisms underlie the sex-specific associations between testosterone and NAFLD. Some studies have shown that men with simple steatosis or advanced NAFLD had lower testosterone levels than healthy controls of the same age [12,54]. Moreover, a recent study found that testosterone treatment is associated with reduced adipose tissue dysfunction and NAFLD in obese hypogonadal men, which suggests that testosterone may play a protective role in the progression of NAFLD and improves NAFLD by reducing intrahepatic triglycerides content [55]. While testosterone formulations are approved for men, they are not recommended for women [56]. Currently, most studies linking testosterone and NAFLD are based on cohorts of women with polycystic ovary syndrome (PCOS), which is an endocrinopathy typically marked by metabolic comorbidities and high androgen levels, such as testosterone. Women with PCOS are a specific high-risk group for NAFLD [57], and testosterone has been shown to increase their risk of NAFLD, independent of obesity and insulin resistance [58]. The results were consistent with our findings from the meta-analysis, men with NAFLD had lower testosterone levels, while women with NAFLD had higher testosterone levels compared to non-NAFLD controls.

Interestingly, we observed in our cohort that women with NAFLD had lower testosterone levels than those without NAFLD, while in the meta-analysis, women with NAFLD had higher testosterone levels than those without. In the study performed by Yim et al., NAFLD was diagnosed by ultrasound fatty liver index and liver enzymes. Among the same number of participants, the overall prevalence of suspected NAFLD was different (i.e., 27.9% for men based on liver enzyme elevation and 29.6% for men based on the ultrasound fatty liver index). We included twice the Yim et al. study based on different diagnostic methods and with the different OR and CI, so it is displayed with different weights. We only included seven studies on testosterone and NAFLD for women in the meta-analysis, further research should investigate the levels of testosterone that may contrition to the development of NAFLD.

### 4.3. SHBG

SHBG has traditionally been considered to function as a transporter of sex steroids, controlling circulating free hormone concentrations [59]. Consistently, all included studies on SHBG with NAFLD indicated that people with NAFLD had lower SHBG levels compared to non-NAFLD controls. Moreover, the peroxisome-proliferator receptors (PPARs) are nuclear fatty acid receptors and act as metabolic sensors and regulators of lipid and glucose homeostasis in many cell types, including the liver. The human SHBG promoter comprises a PPAR-response element. Previous data have shown that PPARγ (one of the PPARs family members) can repress SHBG expression in liver cells [60]. Currently, there is no FDA-approved treatment for NAFLD or NASH, but PPAR agonists improve metabolic dysfunction, inflammation, and oxidative stress which are associated with these liver diseases [61]. In metabolic dysfunctions like NAFLD and NASH, there is already a compensatory increase of PPARγ to counteract SHBG repression. Probably the endogenous level is bound to a maximum, such that for a further increase, an external PPARγ antagonism is needed (e.g., pioglitazone tablets) to further counteract SHBG repression.

### 4.4. DHEA and DHEAS

A study on animal models has shown that the protective effect of DHEA on high-fat-induced hepatic glycolipid metabolic disorder and insulin resistance might be achieved through the activation of the *AMPK-PGC-1α-NRF-1* and *IRS1-AKT-GLUT2* signaling pathways [62]. Another study based on 439 patients with NAFLD (78 in an initial and 361 invalidation cohorts) and in controls with cholestatic liver disease (*n* = 44), found that more advanced NAFLD, as indicated by the presence of NASH with advanced fibrosis stage, is strongly associated with low circulating levels of DHEAS [19]. In the current meta-analysis, we only included one study on DHEA or DHEAS on NAFLD, therefore, further studies may need to investigate the association of DHEA and DHEAS with NAFLD.

### 4.5. Strengths and Limitations

Our study has several strengths. First, it is embedded within the Rotterdam Study which has a prospective design and we included a comprehensive assessment of sex steroids and SHBG. Second, we were able to perform adequate adjustments for a broad range of possible confounders. We also conducted several sensitivity analyses and investigated the association of sex steroids and SHBG with not only NAFLD but also MAFLD (the new definition of NAFLD). Additionally, our study used the LC-MS/MS method to measure serum total testosterone levels that are specific, accurate, and suitable in routine clinical practice and offer advantages over immunoassays regarding lack of interference [63]. Third, our study included in addition to the analysis of primary data, a systematic review and meta-analysis of available published observational studies, and most of the studies included in our meta-analysis were adjusted for potential confounders.

However, some limitations of our study need to be addressed here. On one hand, most eligible studies had cross-sectional study designs, as such, the ability to assess causality or temporality was limited. Moreover, in the Rotterdam Study, total estradiol was measured by using an immunoassay with a detection limit of 18.35 pmol/L, which is considered suboptimal, especially in men and postmenopausal women. To address this issue, we performed logistic regression by estradiol tertiles and estradiol continuous data, which provided similar results. On the other hand, most of the included studies in the meta-analysis used ultrasonography to diagnose NAFLD. Although this imaging technique is widely accepted as the diagnostic method of screening NAFLD due to its low cost, safety, and accessibility, it has limited accuracy in detecting mild steatosis and operator dependency. Finally, significant heterogeneity was observed in the meta-analysis and the possible sources of these heterogeneities include the differences in the continents, methodology, study size, and study design, as demonstrated by the subgroup analyses, and the biggest changes in the recalculated OR and I^2^ resulted when leaving out the N. Wang et al. study by the leave-one-out sensitivity analysis. The study size was the most important factor among these factors by multivariable meta-regression.

## 5. Conclusions

In this study, we observed that lower SHBG levels were associated with NAFLD in both sexes. In addition, men with NAFLD had lower levels of testosterone, and women with NAFLD had higher testosterone levels than non-NAFLD controls. Moreover, our results showed that estradiol may play a potential protective role in the progression of NAFLD among healthy men. Future research should involve larger sample sizes and longitudinal designs to confirm the association of sex steroids and SHBG with NAFLD and uncover underlying molecular mechanisms by which sex hormones may protect against NAFLD. Inconsistent findings across different studies included in this study may be a consequence of variations in study protocols (e.g., differences in sex steroids measurement, methods to diagnose NAFLD, and the study design) and the characteristics of participants (including various comorbidities, genetic susceptibility, and years since menopause onset among women). Further well-designed studies are needed to clarify which type and dose of hormone medication used may affect the sex hormone levels, time since menopause onset in women, metabolic status, and standardized methods to assess sex hormones and NAFLD to account for investigating the associations between sex steroids, SHBG, and NAFLD.

## Figures and Tables

**Figure 1 genes-13-00966-f001:**
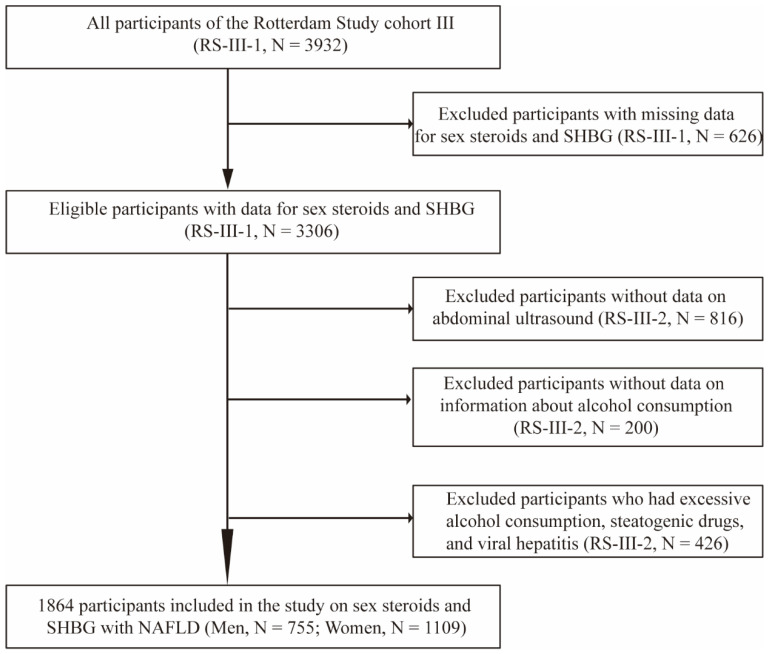
Flow diagram of selection of study participants in the Rotterdam Study. Abbreviations: RS, Rotterdam Study; SHBG, sex hormone-binding globulin; RS-III-1, the first visit of the third cohort; RS-III-2, the second visit of the third cohort; NAFLD, non-alcoholic fatty liver disease.

**Figure 2 genes-13-00966-f002:**
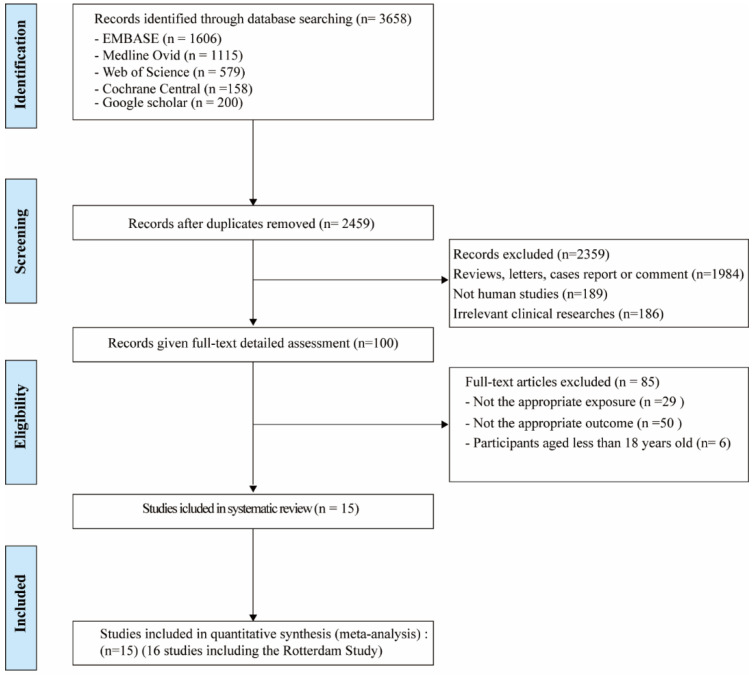
Flow chart of studies included in the systematic review and meta-analysis.

**Figure 3 genes-13-00966-f003:**
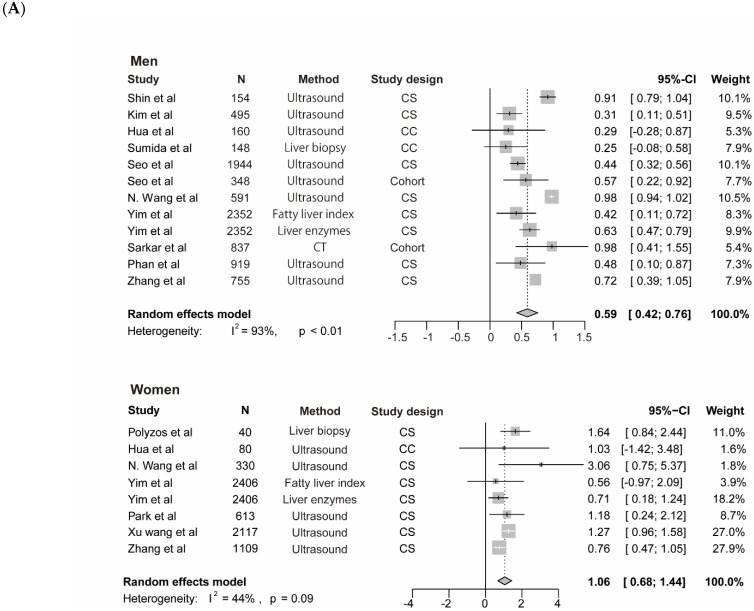
Association of testosterone with NAFLD and results of multivariable meta-regression analysis. (**A**) association of testosterone with NAFLD among men and women. The summary estimate (center of diamond) and 95% confidence interval (width of diamond) were synthesized by using a random effect model which is shown in bold. (**B**) the results of multivariable meta-regression analysis based on included studies for analysis of testosterone and NAFLD among men. Abbreviations: NAFLD, non-alcoholic fatty liver disease; CT, computed tomography; CS, cross-sectional; CC, case-control.

**Figure 4 genes-13-00966-f004:**
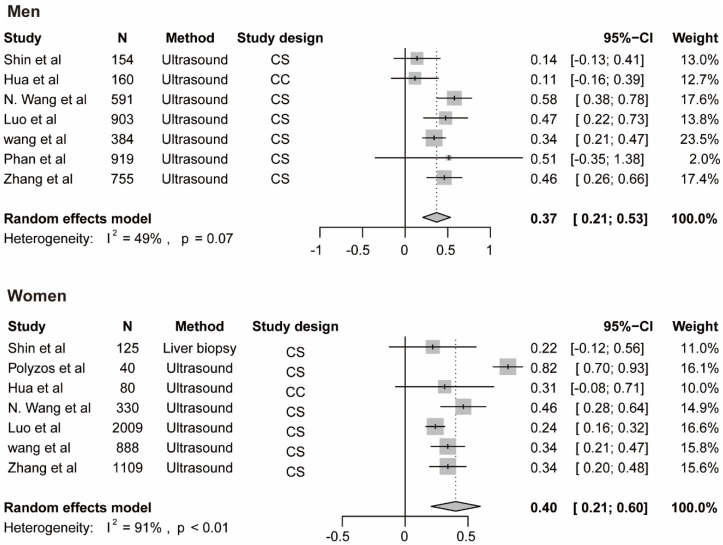
Association of SHBG with NAFLD. The summary estimate (center of diamond) and 95% confidence interval (width of diamond) were synthesized by using a random effect model which is shown in bold. Abbreviations: NAFLD, non-alcoholic fatty liver disease; SHBG, sex hormone-binding globulin.

**Table 1 genes-13-00966-t001:** Characteristics of the study population in the Rotterdam Study.

	Men (*n* = 755)		Women (*n* = 1109)	
	Non-NAFLD (*n* = 482)	NAFLD (*n* = 273)	Non-NAFLD (*n* = 753)	NAFLD (*n* = 356)
Age, years	56.3 ± 6.3	56.8 ± 5.6	56.3 ± 6.0	56.8 ± 6.2
BMI, kg/m^2^	26.4 ± 3.0	29.5 ± 3.9 ***	26.3 ± 4.3	30.7 ± 5.1 ***
Waist circumference, cm	95.5 ± 9.3	105.1 ± 10.9 ***	86.2 ± 11.5	98.4 ± 12.0 ***
Ever smoking, *n* (%)	79 (16.4)	51 (18.7)	159 (21.1)	56 (15.7) *
Alcohol consumption, g/day	6.4 (1.6–8.6)	6.4 (1.6–15.0)	1.6 (0.5–8.6)	1.6 (0.5–6.4)
Hypertension, *n* (%)	230 (47.7)	160 (58.6) **	271 (36.0)	220 (61.8) ***
Type 2 diabetes, *n* (%)	51 (10.6)	74 (27.1) ***	67 (8.9)	50 (14.0) **
HDL, mmol/L	1.2 (1.1–1.4)	1.1 (0.9–1.3) ***	1.6 (1.3–1.9)	1.4 (1.2–1.6) ***
Triglycerides, mmol/L	1.3 (1.0–1.8)	1.5 (1.1–2.2) ***	1.2 (0.8–1.4)	1.4 (1.1–2.0) ***
Total estradiol, pmol/L	92.8 (74.9–118.9)	95.4 (75.8–115.6)	30.2 (18.4–68.0)	36.3 (18.4–71.5)
Total testosterone, nmol/L	17.9 (14.5–22.1)	15.1 (12.0–18.8) ***	0.8 (0.6–1.1)	0.7 (0.5–1.0) ***
SHBG, nmol/L	43.7 (34.7–53.3)	35.1 (27.6–45.2) ***	66.1 (49.1–91.1)	43.5 (31.7–61.3) ***
DHEA, nmol/L	13.6 (8.6–19.1)	11.7 (7.5–16.5) **	13.4 (8.9–19.6)	12.8 (8.6–19.1) *
DHEAS, nmol/L	3585.5 (2513.1–4919.2)	3497.3 (2210.6–4852.8)	2380.0 (1479.3–3451.2)	2317.9 (1455.7–3354.7)
Using hormone medication, *n* (%)	1 (0.2)	1 (0.4)	57 (7.6)	21 (5.9)
Using lipid-lowering medication, *n* (%)	100 (20.7)	85 (31.1) **	136 (18.1)	87 (24.4) *
Postmenopausal women, *n* (%)	NA	NA	407 (54.1)	186 (52.2)
Time since menopause, years #	NA	NA	7.5 (4.3–12.5)	7.7 (4.4–12.4)

Values are presented as a number (percentage) for categorical variables, and the mean (standard deviation) or median (25th–75th quartile) for continuous variables. # Time since menopause is based on 584 women. Non-NAFLD versus NAFLD, * *p*-value <0.05, ** *p*-value <0.01, *** *p*-value < 0.001. Abbreviations: NAFLD, non-alcoholic fatty liver disease; BMI, body mass index; HDL, high-density lipid; SHBG, sex hormone-binding globulin; DHEA, Dehydroepiandrosterone; DHEAS, Dehydroepianhdrosterone sulfate; NA, not applicable.

**Table 2 genes-13-00966-t002:** Associations of sex steroids and SHBG with NAFLD among men.

	Total Estradiol			Continuous	*p* Trend
	Tertile 1	Tertile 2	Tertile 3		
Case subjects	88	97	88		
Model 1, OR (95%CI)	1 (Reference)	1.15 (0.77–1.72)	0.81 (0.53–1.23)	0.84 (0.69–1.02)	0.09
Model 2, OR (95%CI)	1 (Reference)	1.20 (0.79–1.82)	0.84 (0.54–1.29)	0.87 (0.72–1.06)	0.17
	Total testosterone			Continuous	*p* trend
	Tertile 1	Tertile 2	Tertile 3		
Case subjects	122	90	61		
Model 1, OR (95%CI)	1 (Reference)	0.81 (0.54–1.20)	**0.57 (0.37–0.86) ****	0.80 (0.65–0.99) *	0.036
Model 2, OR (95%CI)	1 (Reference)	0.93 (0.62–1.34)	0.72 (0.47–1.12)	0.92 (0.74–1.13)	0.41
	SHBG			Continuous	*p* trend
	Tertile 1	Tertile 2	Tertile 3		
Case subjects	132	80	61		
Model 1, OR (95%CI)	1 (Reference)	**0.51 (0.34–0.75) *****	**0.37 (0.24–0.56) *****	**0.63 (0.52–0.76) *****	1.56 × 10^−6^
Model 2, OR (95%CI)	1 (Reference)	0.60 (0.40–0.90) *	**0.46 (0.30–0.71) *****	**0.72 (0.59–0.88) ****	0.0012
	DHEA			Continuous	*p* trend
	Tertile 1	Tertile 2	Tertile 3		
Case subjects	106	92	75		
Model 1, OR (95%CI)	1 (Reference)	1.05 (0.70–1.59)	0.65 (0.42–1.00) *	0.87 (0.76–1.00) *	0.043
Model 2, OR (95%CI)	1 (Reference)	1.04 (0.68–1.60)	0.72 (0.46–1.11)	0.91 (0.79–1.04)	0.15
	DHEAS			Continuous	*p* trend
	Tertile 1	Tertile 2	Tertile 3		
Case subjects	97	89	87		
Model 1, OR (95%CI)	1 (Reference)	0.97 (0.64–1.46)	1.08 (0.71–1.66)	0.97 (0.89–1.05)	0.39
Model 2, OR (95%CI)	1 (Reference)	0.95 (0.62–1.46)	1.17 (0.76–1.81)	0.98 (0.91–1.06)	0.61

The *p*-value surpassing the significance threshold (*p*-value < 0.05). * *p*-value < 0.05, ** *p*-value < 0.01, *** *p*-value < 0.001. Associations that remain significant at a Bonferroni corrected *p*-value < 0.01 for 5 tests are indicated in bold. Model 1: age+ time difference between hormone measurement and performed ultrasound + BMI + never smoking + alcohol consumption; Model 2: Model 1 + hypertension + T2D + HDL + triglycerides. Abbreviations: OR, odds ratio; CI, confidence interval; SHBG, sex hormone-binding globulin; DHEA, Dehydroepiandrosterone; DHEAS, Dehydroepiandrosterone sulfate; BMI, body mass index; T2D, type 2 diabetes; HDL, high density lipid.

**Table 3 genes-13-00966-t003:** Associations of sex steroids and SHBG with NAFLD among women.

	Total Estradiol			Continuous	*p* Trend
	Tertile 1	Tertile 2	Tertile 3		
Case subjects	102	121	133		
Model 1, OR (95%CI)	1 (Reference)	1.20 (0.84–1.70)	1.21 (0.85–1.72)	1.00 (0.95–1.06)	0.82
Model 2, OR (95%CI)	1 (Reference)	1.21 (0.83–1.78)	1.18 (0.80–1.73)	1.01 (0.96–1.06)	0.79
	Total testosterone			Continuous	*p* trend
	Tertile 1	Tertile 2	Tertile 3		
Case subjects	130	125	101		
Model 1, OR (95%CI)	1 (Reference)	0.91 (0.65–1.28)	0.68 (0.48–0.97) *	**0.81 (0.71–0.92) ****	0.001
Model 2, OR (95%CI)	1 (Reference)	0.98 (0.68–1.40)	0.76 (0.52–1.11)	0.85 (0.76–0.96) *	0.011
	SHBG			Continuous	*p* trend
	Tertile 1	Tertile 2	Tertile 3		
Case subjects	202	96	58		
Model 1, OR (95%CI)	1 (Reference)	**0.39 (0.28–0.54) *****	**0.27 (0.18–0.38) *****	**0.60 (0.54–0.68) *****	2.23 × 10^−16^
Model 2, OR (95%CI)	1 (Reference)	**0.50 (0.34–0.71) *****	**0.34 (0.23–0.52) *****	**0.69 (0.61–0.78) *****	1.20 × 10^−9^
	DHEA			Continuous	*p* trend
	Tertile 1	Tertile 2	Tertile 3		
Case subjects	122	125	109		
Model 1, OR (95%CI)	1 (Reference)	1.09 (0.78–1.54)	1.02 (0.72–1.44)	0.98 (0.89–1.08)	0.71
Model 2, OR (95%CI)	1 (Reference)	1.12 (0.78–1.61)	1.07 (0.73–1.56)	1.01 (0.92–1.11)	0.88
	DHEAS			Continuous	*p* trend
	Tertile 1	Tertile 2	Tertile 3		
Case subjects	126	115	115		
Model 1, OR (95%CI)	1 (Reference)	1.01 (0.72–1.43)	0.94 (0.66–1.34)	0.99 (0.90–1.09)	0.86
Model 2, OR (95%CI)	1 (Reference)	1.03 (0.71–1.50)	0.98 (0.67–1.44)	1.01 (0.92–1.10)	0.84

The *p*-value surpassing the significance threshold (*p*-value < 0.05). * *p*-value < 0.05, ** *p*-value < 0.01, *** *p*-value < 0.001. Associations that remain significant at a Bonferroni corrected *p*-value < 0.01 for 5 tests are indicated in bold. Model 1: age+ time difference between hormone measurement and performed ultrasound + BMI + ever smoking + alcohol consumption; Model 2: Model 1 + hypertension + T2D + HDL + triglycerides + postmenopausal status. Abbreviations: OR, odds ratio; CI, confidence interval; SHBG, sex hormone-binding globulin; DHEA, Dehydroepiandrosterone; DHEAS, Dehydroepiandrosterone sulfate; BMI, body mass index; T2D, type 2 diabetes; HDL, high density lipid.

**Table 4 genes-13-00966-t004:** Characteristics of the included studies for meta-analysis.

Lead Author Publication Date	Location	Average Age/Age Range (years)	Study Design	Number of Participants	Exposure	Outcome, Measurement Method	Covariates Adjusted For	Study Quality *
Shin et al., 2011 [35]	Korea	57.0	Cross-sectional	Men 154 Women 125	Testosterone and SHBG	NAFLD, Ultrasound	Age, BMI, waist circumference, hypertension, TG, ALT, γGT, CRP, HOMA-IR, estradiol, total testosterone, and antidiabetic medications	6
Tian et al., 2012 [36]	China	20–60	Cross-sectional	Men 1882	Estradiol	NAFLD, Ultrasound	None	6
Kim et al., 2012 [12]	Korea	54.4	Cross-sectional	Men 495	Testosterone	NAFLD, Ultrasound	Age, smoking, diabetes, exercise, BMI, TG, HDL cholesterol, HOMA-IR, hs-CRP, and VAT	7
Polyzos et al., 2013 [37]	Greece	55.7	Cross-sectional	Women 40		NAFLD and NASH, Liver biopsy	Age, BMI, and waist circumference	7
Hua et al., 2014 [46]	China	56.8	Case-control	Men 160, Women 80	Total testosterone and SHBG	NAFLD, Ultrasound	Age, smoking status, alcohol use, diabetes, BMI, fasting C-peptide	7
Sumida et al., 2015 [38]	Japan	57.0	Case-control	Men 148	Free testosterone	NAFLD, Liver biopsy	NA	6
Seo et al., 2015 [39]	Korea	38.0–54.0	Cohort	Men 1944	Testosterone	NAFLD, Ultrasound	Age, smoking, exercise, history of hypertension and diabetes, systolic blood pressure, glucose, HDL cholesterol, TG, and ESR	8
N. Wang et al., 2016 [14]	China	57.0	Cross-sectional	Men 2689, Women 1461	Testosterone and SHBG	Mild NAFLD and moderate-severe NAFLD, Ultrasound	Age, total testosterone, abdominal obesity, diabetes, LDL cholesterol, HDL cholesterol, TG, and systolic blood pressure	7
Yim et al., 2017 [40]	United States	47.0	Cross-sectional	Men 2352 Women 2406	Testosterone	NAFLD, Serum alanine aminotransferase	Age, BMI, ethnicity, education level, marital status, economic status, the presence of hypertension, and total cholesterol	8
Luo et al., 2018 [41]	China	40–75	Cross-sectional	Men 903 Women 2009	SHBG	NAFLD, Ultrasound	Age, sex, postmenopausal status, household income, WHR, truck fact, current smoking and drinking, physical activity, hypertension, diabetes, serum glucose, HOMA-IR, TG, HDL cholesterol, ALT, UA, testosterone, and DHEAS levels	7
Park et al., 2019 [42]	Korea	21–75	Cross-sectional	Women 613	Testosterone	NAFLD, Ultrasound	Age, regular exercise, type 2 diabetes, BMI, MAP, FPG, TG, HDL cholesterol, and testosterone levels	7
Sarkar et al., 2019 [47]	United States	35.1	Cohort	Men 837	SHBG	NAFLD, Computed tomography	Age, race, BMI, waist circumference, LDL cholesterol, TG, and HOMA-IR	8
Wang et al., 2019 [43]	China	60.4	Cross-sectional and cohort	Men 384, Women 888	SHBG	NAFLD, Ultrasound	Age, gender, and postmenopausal status, BMI, WHR, trunk fat mass; physical activities; current smoking and drinking; history of hypertension and diabetes; HOMA-IR, TG, LDL cholesterol /HDL cholesterol, UA, albumin, ALP, DHEAS	8
Phan et al., 2020 [45]	United States	40.4	Cross-sectional	Men 919	Estradiol, testosterone, and SHBG	NAFLD, Ultrasound	Age, race, smoking, alcohol, physical activity, waist circumference	8
Xu Wang et al., 2021 [44]	China	59.83	Cross-sectional	Women 2117	Testosterone	NAFLD, Ultrasound	Age, postmenopausal status, body mass index, waist-to-hip ratio, physical activity, smoking, hypertension, diabetes, dyslipidemia, triglycerides, total cholesterol, CRP	8
Zhang et al., Current study	Netherlands	55.9	Cross-sectional	Men 869, Women 1128	Estradiol, testosterone, SHBG, DHEA, and DHEAS	NAFLD, Ultrasound	Age, sex, time difference between hormone measurement and performed ultrasound, BMI, ever smoking, alcohol consumption, hypertension, T2D, HDL, triglyceride, total cholesterol	8

* Quality assessment based on the Newcastle-Ottawa Scale. Range 0 to 9, a higher score is higher quality. Abbreviations: FLD, fatty liver disease; ALD, alcoholic liver disease; NA, not available; NAFLD, non-alcoholic fatty liver disease; NASH, Non-alcoholic steatohepatitis; MAFLD, metabolic dysfunction-associated fatty liver disease; BMI, body mass index; LDL, low-density lipoprotein; TG, triglycerides; HOMA-IR, homeostatic model assessment for insulin resistance; WHR, waist-to-hip ratio; ALP, alkaline phosphatase; DHEAS, testosterone, and dehydroepiandrosterone sulfate levels; HDL, high-density lipoprotein; UA, uric acid; ALT, alanine aminotransferase; MAP, mean arterial pressure; FPG, fasting plasma glucose; γGT, γ-glutamyl transpeptidase; CRP, C-reactive protein; hs-CRP, high-sensitivity C-reactive protein; VAT, variance inflation factors; ESR, erythrocyte sedimentation rate. FBG, fasting blood glucose; HBeAg, hepatitis B e antigen.

## Data Availability

Data can be obtained upon request. Requests should be directed towards the management team of the Rotterdam Study (secretariat.epi@erasmusmc.nl), which has a protocol for approving data requests. Due to restrictions based on privacy regulations and informed consent of the participants, data cannot be made freely available in a public repository.

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
