# Peer review of "Associations of Sex Steroids and Sex Hormone-Binding Globulin with Non-Alcoholic Fatty Liver Disease: A Population-Based Study and Meta-Analysis"

_genes, 2022, doi:10.3390/genes13060966_

Round 1

Reviewer 1 Report

From my point of view, I could not find any substantial flaws in this study. It is believed that even if the decision to publish is decided after a solution to the questions about the simple matters below.   1. In the recent meta-analyis implementation, it is usually customary to register a protocol for conducting research in advance. Could the author explain why they did not pre-register the systematic review of this study in PROSEPRO or other databases?   2. Looking at Figure 3, it is thought that both male and female study subjects of 'Yim et al' were included twice. Although the research method is indicated differently (Fatty liver index and Liver enzyme), it seems necessary to further explain how the number of the same subjects is displayed with different weights.   3. The heterogeneity of the meta-analysis results supporting “The pooled ORs for NAFLD comparing T3 versus T1 of testosterone were 0.59” is too large. Due to the design characteristics of this study, it is considered that it will be more convincing if sensitivity analysis according to the leave-one-out method is additionally performed in addition to subgroup analysis.   4. For readability, wouldn't it be better to move the table on the characteristics of the 'observational studies' to be analyzed included in the meta-analysis to the body rather than the supplementary file?   I hope my comments have been helpful to the Journal and its authors. thank you.  

Author Response

Response to Reviewer 1 Comments

From my point of view, I could not find any substantial flaws in this study. It is believed that even if the decision to publish is decided after a solution to the questions about the simple matters below.

1. In the recent meta-analysis implementation, it is usually customary to register a protocol for conducting research in advance. Could the author explain why they did not pre-register the systematic review of this study in PROSEPRO or other databases?

Response 1. We appreciate the thoughtful comment of the reviewer. The current meta-analysis, despite was not registered in online platforms, was conducted in line with the published approach how to do a systematic review and meta-analysis (Muka et al. 2020) and using a predefined protocol available among authors. We have now commented on this in materials and methods section.

Lines 135-138: The study was conducted using a predesigned protocol (which was not registered on online platforms) and following a published guideline on how to perform a systematic re-view and metaanalysis,27 and was reported in accordance with PRISMA guidelines (Supplementary Table 4).

2. Looking at Figure 3, it is thought that both male and female study subjects of 'Yim et al' were included twice. Although the research method is indicated differently (Fatty liver index and Liver enzyme), it seems necessary to further explain how the number of the same subjects is displayed with different weights.

Response 2. We acknowledge the reviewer’s comment and agree that it should be further explained how the number of the same subjects is displayed with different weights. The weight of a study depends on the number of participants and events in the study. The overall sample size is not the only determining factor. The value of the point estimate and range of the confidence interval (CI) are clues to the weight of a study. In the ‘Yim et al.’ study, among the same subjects, the prevalence of NAFLD is different based on the fatty liver index and liver enzymes. Thus, different odds ratio and CI among the same number of subjects was displayed with different weight. We have now clarified the reason for different weights among the same subjects.

Lines 415-420: In the study performed by Yim et al., NAFLD was diagnosed by ultrasound fatty liver index and liver enzymes. Among the same number of participants, the overall prevalence of suspected NAFLD was different (such as 27.9% for men based on liver enzymes elevation and 29.6% for men based on the ultrasound fatty liver index). We included twice “Yim et al.” based on different diagnostic methods and with the different OR and CI, so it is displayed with different weights.

3. The heterogeneity of the meta-analysis results supporting “The pooled ORs for NAFLD comparing T3 versus T1 of testosterone were 0.59” is too large. Due to the design characteristics of this study, it is considered that it will be more convincing if sensitivity analysis according to the leave-one-out method is additionally performed in addition to subgroup analysis.

Response 3. We thank the reviewer for this suggestion. We performed the leave-one-out sensitivity analysis and the results are shown in the updated supplementary files (Supplementary figure 2). We have now revised the manuscript accordingly.

Supplementary figure 2. Leave-one-out sensitivity analysis for the meta-analysis of testosterone and NAFLD among men. The left plot is sorted by effect size and the right plot is sorted by the heterogeneity I2. Abbreviations: NAFLD, non-alcoholic fatty liver disease; FLI, fatty liver index; CS, cross-sectional.

Lines 197-200: For the leave-one-out sensitivity analysis, we used the R package “metafor”, “dmetar”, and “InfluenceAnalysis” function to recalculate the results of the meta-analysis N times (N= the number of the included studies for meta-analysis), each time leaving out one study.

 Lines 340-343: In addition, we performed the leave-one-out sensitivity analysis of included studies of testosterone and NAFLD among men. The biggest changes in the recalculated OR and I2 when leaving out the N. Wang et al. study (OR: 0.55, 95%CI 0.39-0.70, I2 =80%), the details are shown in Supplementary figure 2.

4. For readability, wouldn't it be better to move the table on the characteristics of the 'observational studies' to be analyzed included in the meta-analysis to the body rather than the supplementary file? I hope my comments have been helpful to the Journal and its authors. thank you.

 Response 4. As suggested by the reviewer, we have now moved the table on the characteristics of the included studies to the main text, Table 4 in the revised manuscript.

Reviewer 2 Report

In the manuscript "Associations of sex steroids and sex hormone-binding globulin with non-alcoholic fatty liver disease: a population-based study and meta-analysis" Xiaofang Zhang has reviewed the associations of NAFLD with sex steroids and sex hormone-binding globulin (SHBG) in the population-based study and have performed a meta-analysis. the authors have reviewed the literature well and have presented a comprehensive report of the current literature and have also presented their meta-analysis in an easy-to-understand way. apart from the few typos and English language corrections the manuscript looks good and would be of interest to the reader.

Author Response

Response to Reviewer 2 Comments

In the manuscript "Associations of sex steroids and sex hormone-binding globulin with non-alcoholic fatty liver disease: a population-based study and meta-analysis" Xiaofang Zhang has reviewed the associations of NAFLD with sex steroids and sex hormone-binding globulin (SHBG) in the population-based study and have performed a meta-analysis. the authors have reviewed the literature well and have presented a comprehensive report of the current literature and have also presented their meta-analysis in an easy-to-understand way. apart from the few typos and English language corrections the manuscript looks good and would be of interest to the reader.

Response. We highly appreciate the reviewers' supportive comments on our manuscript. We have checked and corrected the typos and English language, and the changes in the updated manuscript are highlighted with “Track changes”.
